# The Effect of Breast Milk Microbiota on the Composition of Infant Gut Microbiota: A Cohort Study

**DOI:** 10.3390/nu14245397

**Published:** 2022-12-19

**Authors:** Yapeng Li, Lei Ren, Yunyi Wang, Jinxing Li, Qingqing Zhou, Chenrui Peng, Yuchen Li, Ruyue Cheng, Fang He, Xi Shen

**Affiliations:** 1Department of Nutrition and Food Hygiene, West China School of Public Health, West China Fourth Hospital, Sichuan University, Chengdu 610041, China; 2Hebei Inatural Bio-Tech Co., Ltd., Shijiazhuang 050899, China; 3Department of Nutrition, Sichuan Academy of Medical Sciences•Sichuan Provincial People’s Hospital, Chengdu 610072, China; 4Mental Health Center, West China Hospital of Sichuan University, Chengdu 610044, China; 5West China Biomedical Big Data Center, West China Hospital of Sichuan University, Chengdu 610044, China

**Keywords:** infant, gut microbiota, breastfeeding, SCFAs

## Abstract

Evidence shows that breast milk microbiota and an infant’s gut microbiota are related. This study aimed to compare the effects of breast milk microbiota on the construction and colonization of gut microbiota in newborns. In this study, 23 healthy infants were selected and divided into a breastfeeding group (13) and a mixed feeding group (10) based on the feeding method within one month of age. Infant fecal and breast milk samples were collected on the day of birth (0 day) and 30 days after birth (30 days) for 16S rRNA second-generation sequencing and SCFA detection. The results showed that *Firmicutes* and *Actinobacteriota* on day 0 and *Firmicutes* and *Proteobacteria* on 30 d dominated breast milk gut microbiota. There were correlations between the breast milk microbiota and the infant gut microbiota in each group (*p* < 0.05). Additionally, breast milk microbiota correlated more significantly with infants’ SCFAs in the breastfeeding group than in the mixed feeding group. This study showed that breast milk microbiota partially influences the construction of infant gut microbiota, with some key strains having a crucial influence, such as *Lactobacillus*, *Bifidobacterium*, and *Enterobacter*. However, the effect of breast milk microbiota on infant gut microbiota is not through direct strain transmission but has been indirectly influenced, which may be related to the cross-feeding effect mediated by SCFAs.

## 1. Introduction

Early life is a critical period for colonizing and constructing the gut microbiota. During this time, the infant gut microbiota is dynamically constructed and influenced by several factors, such as delivery mode, feeding mode, and so on [1,2]. Although many factors influence gut microbiota, the origin of infant gut microbiota remains debatable. Previous studies have suggested that the origin of infant gut microbiota is primarily the migration of maternal intestinal and vaginal microbiota during delivery, and the delivery mode can influence the development of the infant gut microbiota up to six months of age or beyond [3]. However, it has been suggested that the contact between the infant’s and mother’s skin during breastfeeding leads to microorganism migration into the newborn’s mouth and intestine [4]. As research on gut microbiota progressed, a breakthrough was made in the controversy over the origin of the infant gut microbiota. In 1984, it was reported that strictly anaerobic bifidobacteria could be detected in breast milk [5], suggesting that breast milk is unsterile and that the microorganisms in it are not entirely caused by contamination as the breast was cleaned before collection [5,6]. In conclusion, no matter the delivery mode, skin or breast milk microbiota influence the infant gut microbiota, but the primary origin remains unclear. Under these circumstances, some studies have proposed the “Gut-Lactation Pathway”, which means bacteria from the mother’s intestine may migrate to breast milk through the endogenous pathway. These bacteria further migrate into the mouth and intestine of the newborn, which may contribute to the maturation of the infant’s gut microbiota and immune system. Above all, the origin of the infant gut microbiota is widely discussed. However, the specific way each origin affects the infant gut microbiota is unclear (including the delivery mode, skin microbiota, and breast milk microbiota). The origin may be the direct migration of certain bacteria or the indirect influence of certain bacteria. Therefore, this study focuses on the breast milk pathways and the specific effects of breast milk microbiota on the infant’s gut microbiota.

Generally, breast milk provides optimal nutrition for infants in the first six-months of life. The feeding method during the neonatal period affects their nutritional support in early life and is a key factor in constructing their gut microbiota. Breastfeeding may affect short-chain fatty acids (SCFAs), the metabolites of the infant’s intestinal microbiota, and conversely, SCFAs may also affect the infant’s intestinal instead [7]. Currently, in existing mother-infant cohorts, although feeding methods affect the infant gut microbiota, the correlation between breast milk microbiota and the dynamic construction of the infant gut microbiota is unclear. This study aimed to compare the effect of breast milk microbiota on the construction and colonization of infant gut microbiota and the physiologically active substances metabolized by the gut microbiota. Additionally, the unique role of breast milk microbiota in the construction of the infant gut microbiota was further explored.

## 2. Materials and Methods

### 2.1. Subjects Selection

For this study, 23 pairs of healthy full-term newborns born between January 2020 and August 2020 at the West China Second University Hospital and their mothers were selected. Infants with no congenital defect-type or genetic diseases, no acute diseases, such as neonatal jaundice, and normal weight were included. Infants were excluded if their parents had chronic diseases or if their mothers had taken antibiotics, probiotics, or prebiotic products within one month before delivery. Moreover, infants would also be excluded if they developed a severe disease during the follow-up process.

Subjects were divided into the breastfeeding group (BF) and the mixed feeding group (MF). The breastfeeding group (BF) comprised 13 pairs of newborns who were breastfed for ≥25 days within 30 days after birth. Since the newborns would not have been fed only formula, they were more or less breastfed. The remaining ten infants who were formula-fed for ≥25 days within 30 days of birth belonged to the MF. All infants’ parents voluntarily participated in this study and signed an informed consent form.

At the initial study visit, personnel collected newborn physical measurements from hospital records. Mothers were asked to complete a questionnaire for information, including the health status of newborns (the frequency of diarrhea and the prevalence of allergic diseases), drug use (mainly the time, frequency, and dosage of antibiotic drugs), and diet (especially the use of probiotics, prebiotics, yogurt, and other products).

The trial followed the Helsinki principles and was ethically reviewed and approved by the Ethics Committee of the West China School of Public Health/the Fourth West China Hospital of Sichuan University (K2020039). An informed consent form was signed by the mother (guardian) before the trial began.

### 2.2. Sample Collection

Fecal samples were collected from newborns on days 0 and 30 after birth using sterile fecal collection tubes. Breast milk samples were collected from mothers using breast milk bags on days 0 and 30 (the skin around the mother’s nipples was cleaned before breast milk collection). All samples were stored at −20 °C immediately after collection and transferred to a −80 °C refrigerator for measurement within 24 h. A total of 58 fecal samples and 25 breast milk samples were finally collected.

### 2.3. Materials and Reagents

Fecal and breast milk bacterial genomic DNA extraction kit E.Z.N.A. ^®^Stool DNA Kit (D4015, Omega, Inc., Norwalk, CT, USA) (Beijing Tiangen Biochemical Technology Co., Ltd., Beijing, China); 16S rDNA sequencing (Chengdu Bespoke Biotechnology Co., Ltd., Chengdu, China).

### 2.4. DNA Extraction from Fecal and Breast Milk Samples

At each time point, 0.2 g of feces from infants/0.2 mL of breast milk from mothers were taken, and genomic DNA was extracted using DNA extraction kits, respectively. DNA concentration and purity were analyzed using a Nanodrop 2000 ultraviolet-visible microspectrophotometer (Thermo-Fisher Scientific, Inc., Waltham, MA, USA). When DNA concentration > 5 ng/μL and purity (OD260/OD280) between 1.7 and 1.9 were satisfied simultaneously, it indicated that the DNA quality was reliable.

### 2.5. 16S rRNA Sequencing and Bioinformatics Analysis

Here, 16S rRNA sequencing of microorganisms in feces/breast milk was conducted using a second-generation sequencer. The V3–V4 hypervariable region of the 16S rRNA gene was amplified using PCR (338 forward primer 5′-ACTCCTACGGGAG GCAGCAG-3′ and 806 reverse primer 5′-GGACTACHVGGGT WTCTAAT-3′). PCR was conducted using 50 ng of template DNA, 12.5-μL of Phusion^®^ Hot Start Flex 2X Master Mix (NEB, M0536L), 2.5 μL of each of the forward and reverse primers, after which the system volume was adjusted to 25 μL with ddH_2_O. The amplification program was followed by 35 cycles at 98 °C for 30 s, 98 °C for 10 s, 54 °C for 30 s, and 72 °C for 45 s.

PCR products were purified using AMPure XT and quantified using a Qubit kit and library detection. Then, raw sequencing data were preprocessed using bcl2fastq (v1.8.4) and FLASH software (v1.2.11). The operational taxonomic units (OTUs) were clustered against the SILVA 128 reference database at 100% similarity using the USEARCH algorithm. The clustered OTUs were classified into five taxonomic rank categories (phylum, order, class, family, and genus) using the SILVA 128 reference database at 100% similarity. Subsequently, abundance and alpha diversity were calculated for OTUs to analyze the species richness and evenness information within the samples.

### 2.6. Determination of SCFAs in Fecal Samples

Since infant metabolites were unstable at 0 day, this study only measured fecal SCFAs at 30 days in infants. Specific steps are as follows. A 100-mg fecal sample was collected, and the supernatant was taken after pretreatment to determine the fecal SCFAs using gas chromatography-mass spectrometry (GC-MS). Chromatographic conditions: Agilent HP-INNOWAX capillary column (30 m × 0.25 mm × 0.25 μm); split injection, injection volume 1 μL, split ratio 10:1; the inlet was held at 250 °C; carrier gas helium, carrier gas flow rate 1.0 mL/min. The SCFAs content in feces was calculated using the internal standard method with the standard curves of acetic, propionic, and butyric acids.

### 2.7. Statistical Analysis

All statistical analyses were conducted using SPSS 22.0. Data were presented as $\bar{x}$ ± s (means ± standard deviation). The differences between basic information and factors influencing the BF and MF groups were analyzed using the chi-square test or Fisher’s exact test. Normally distributed variables were statistically tested using a two-tailed *t*-test for two independent groups or a one-way analysis of variance for multiple independent groups. Abnormally distributed variables were tested using the Kruskal–Wallis test, and the Nemenyi test was used for multiple comparisons. Spearman rank correlation analysis was used to analyze the relationship between groups, with *p* < 0.05 considered a statistically significant difference. A correlation analysis of the microbiota of breast milk and infants’ guts was conducted, including phylum and genus levels. Several strains of specific beneficial and pathogenic bacteria were selected for further mapping analysis.

## 3. Results

### 3.1. Basic Information about the Population

The sample size and the number of collected samples in each group are described in Table 1. The characteristics of the participants selected for this study are described in Table 2 and Table 3. The mean age of the mothers in the breastfeeding group was 31.77 ± 2.92 years, while the mean age of the MF group was 31.50 ± 5.28 years. Besides, the BF group comprised three boys and ten girls, while the MF group comprised five boys and girls each. No differences were observed in the baseline characteristics using a questionnaire and statistical comparison (*p* > 0.05).

### 3.2. Differences in the Relative Abundance of Breast Milk Microbiota and Infant Gut Microbiota

The breast milk microbiota was analyzed at two time points. The 0 day group (0 day means samples collected on day 0) (*n* = 6) was dominated by the *Firmicutes* and *Actinobacteriota*, while the 30 days group (30 days means samples collected on day 30) (*n* = 19) was dominated by the *Firmicutes* and *Proteobacteria* at the phylum level. Among them, the relative abundance of *Desulfobacterota*, *Campilobacterota,* and *Deferribacterota* was significantly higher on 30 days than on 0 day (*p* < 0.05) (Figure 1A). At the genus level, *Streptococcus* was the dominant genus at both 0 day and 30 days. The relative abundance of the following genus statistically differed between the two groups (*p* < 0.05). Compared with the 0 day group, the relative abundance of *Acinetobacter*, *Pseudomonas*, *Brevundimonas*, and *Serratia* increased, whereas *Veillonella*, *Escherichia-Shigella*, *Bacillus*, *Rothia*, *Gemella*, *Corynebacterium*, *Ammoniphilus*, *Clostridium*, *Listeria*, *Erysipelatoclostridium,* and *Citrobacter* decreased in the 30 days group (Figure 1B). After considering the effects of feeding practices, no statistical difference in breast milk microbiota was observed between the 30-days BF group (*n* = 9) and the 30-days MF group (*n* = 10), as shown in Figure 1C,D.

Similarly, we did the same analysis for the composition of the infant gut microbiota. At the phylum level, the relative abundance of *Actinobacteriota* was significantly higher on 30 days than on 0 day (*p* < 0.05), while at the genus level, the relative abundance of *Veillonella, Bifidobacterium*, and *Clostridium* were significantly higher on 30 days than on 0 day (*p* < 0.05). The figures are performed in the Appendix A.

### 3.3. Correlation between Breast Milk Microbiota and Infant Gut Microbiota

In both groups, correlations were observed between breast milk and infant gut microbiota. At the phylum level, the correlation between breast milk and infant gut microbiota was only reflected in a few specific microbiota in the 0-day BF group, including *Verrucomicrobiota*, *Actinobacteriota*, *Cyanobacteria*, *Chloroflexi*, *Deinococcota*, *Deferribacterota*, *Desulfobacterota*, and *Bacteroidota*. In the 0-day BF group, the presence of *Actinobacteriota* in breast milk was negatively correlated with *Desulfobacterota* in infants’ feces (*r* = −0.894, *p* = 0.041). *Verrucomicrobiota*, *Cyanobacteria*, *Chloroflexi*, *Deinococcota*, and *Bacteroidota* in breast milk was positively correlated with *Actinobacteriota*, *Cyanobacteria*, *Deferribacterota*, and *Desulfobacterota* in infants’ feces, respectively, in the 0-day BF group (*p* < 0.05), as shown in Figure 2A. However, due to the small sample size, the correlation analysis could not be conducted in the MF group at 0 day. On day 30, *Verrucomicrobiota* in the breast milk of the BF group was negatively correlated with *Acidobacteriota* and *Chloroflexi* in infants’ feces (*r* = −0.730, *p* = 0.025; *r* = −0.730, *p* = 0.025). Meanwhile, *Chloroflexi* in breast milk and *Campilobacter* in infants’ feces were positively correlated in the 30-days BF group (*r* = 0.807, *p* = 0.009). The correlation results of the MF group showed that the microbiota in breast milk were positively correlated with the microbiota in infants’ feces (*p* < 0.05). Moreover, it was found that the number of correlations was greater in the MF group than in the BF group (Figure 2B).

At the genus level, many microbiota were correlated within each group. Some common beneficial and pathogenic bacteria and their correlations with other microbiota are listed in Figure 2C–E. In the 0-d BF group (*n* = 5), *Lactobacillus* in breast milk was positively correlated with *Bifidobacterium* (*r* = 1.000, *p* = < 0.001) and *Clostridium* (*r* = 0.900, *p* = 0.037) in the infants’ feces. The presence of *Enterobacter* was positively correlated with *Lactobacillus* (*r* = 0.900, *p* = 0.037) in the infants’ feces. Moreover, *Klebsiella*, a pathogenic bacterium in breast milk, was negatively correlated with *Lactobacillus* (*r* = −1.000, *p* < 0.001) in the infant’s gut, whereas *Escherichia-Shigella* was positively correlated with *Ammoniphilus* (*r* = 0.900, *p* = 0.037) in the *infant’s* gut*. Blautia*, *Rothia*, and *Parabacteroides* in breast milk suggest a positive correlation with pathogenic bacteria found in the infant intestine, such as *Clostridium* and *Escherichia- Shigella* (*p* < 0.05). Furthermore, in the 30-days BF group (*n* = 9), *Lactobacillus* in breast milk was negatively correlated with *Enterobacter* (*r* = −0.783, *p* = 0.013) in the infant’s intestine. *Bifidobacterium* in breast milk was positively correlated with *Alistipes* (*r* = 0.683, *p* = 0.042) and *Muribaculaceae* (*r* = 0.836, *p* = 0.032), and was negatively correlated with *Undefined_Enterobacteriaceae* (*r* = −0.783, *p* = 0.013). *Gemella* in breast milk was negatively correlated with *Bifidobacterium* (*r* = −0.746, *p* = 0.021) in the infant’s intestine. In the 30-days MF group (*n* = 10), the correlation between the key beneficial and pathogenic microbiotas we focused on was reduced compared with the 30-days BF group. Specifically, *Bifidobacterium* in breast milk associated positively with *Enterobacter* (*r* = 0.648, *p* = 0.019) in the infant’s intestine, while *Massilia* associated positively with *Bifidobacterium* (*r* = −0.719, *p* = 0.043) in the infant’s intestine. Moreover, *Pseudomonas* in breast milk was negatively correlated with the pathogenic *Escherichia-Shigella* (*r* = −0.733, *p* = 0.016). All positive and negative results in the three groups were statistically significant and shown in Figure 2F, which indicated that, given the relevant results, the 30-days MF group had the highest number of species correlated between breast milk and infant gut microbiota, while the 30-days BF group had the lowest number of species correlated. Among the focused bacteria, the impact of MF may appear less than that of BF. But overall, among some of the lesser-focused bacteria, MF had additional effects on infant gut microbiota, suggesting that MF causes more uncertainty.

### 3.4. Correlation between Breast Milk Microbiota and Infant Fecal SCFAs

The possible associations between breast milk microbiota and infant fecal SCFAs, including acetic and propionic acids, were analyzed (Figure 3). At both phylum and genus levels, the correlation was more significant in the 30-days BF group (*n* = 5) than in the 30-days MF group (*n* = 5). In the 30-days BF group, *Desulfobacterota*, *Bacteroidota*, and *Proteobacteria* in the breast milk microbiota were negatively correlated with acetic acid in infant feces (*Desulfobacterota*, *r* = −0.900, *p* = 0.037; *Bacteroidota*, *r* = −1.000, *p* < 0.001; *Proteobacteria*, *r* = −0.900, *p* = 0.037). In the 30-days MF group, most of the correlation trends were opposite to those in the BF group, but it was not statistically significant (Figure 3A). At the genus level, the breast milk microbiota of the 30-days BF group showed *Undefined_Rhizobiaceae*, *Granulicatella*, *Undefined_Enterobacteriaceae*, *Serratia*, *Corynebacterium* were negatively correlated with acetic acid in infant feces (*Undefined_Rhizobiaceae*, *r* = −0.894, *p* = 0.041; G*ranulicatella*, *r* = −0.894, *p* = 0.041; *Undefined_Enterobacteriaceae*, *r* = −0.900, *p* = 0.037; *Serratia*, *r* = −0.900, *p* = 0.037; *Corynebacterium*, *r* = −1.000, *p* < 0.001), while *Massilia* and *Citrobacter* were positively correlated with acetic acid (*Massilia*, *r* = 0.900, *p* = <0.001; *Citrobacter*, *r* = 0.900, *p* = 0.037). In the 30-d MF group, *Ammoniphilus* and *Haemophilus* were positively correlated with acetic acid (*Ammoniphilus*, *r* = 0.900, *p* = 0.037; *Haemophilus*, *r* = 0.600, *p* = 0.037), while *Rothia* was positively correlated with propionic acid (*r* = 0.900, *p* = 0.037), as shown in Figure 3B. The concentrations of SCFAs in each group on day 30 are shown in the Appendix A, and none of the results were statistically different (*p* > 0.05).

## 4. Discussion

Infancy is a crucial stage in the rapid evolution of the gut microbiota, which is unstable and susceptible to dysbiosis due to external factors [8]. Various factors influence the establishment of gut microbiota in infancy, including delivery method, feeding method, nutritional status, and antibiotic application [9,10]. However, the origin of the infant gut microbiota is unknown. Furthermore, as strictly anaerobic bifidobacteria have been identified in breast milk, the “Gut-Mammary Pathway” theory has been proposed. To be specific, live bacteria from the maternal intestine travel through the gut-mammary pathway to the mammary gland via an endogenous route. This transfer involves complex interactions between epithelial cells, immune cells, and bacteria [11,12]. Then, these bacteria further migrate into the mouth and intestine of the newborn, probably contributing to the maturation of the infant’s gut microbiota. Therefore, we conjecture that breast milk may be a major origin of the infant gut microbiota after birth. To corroborate this conjecture, this study aims to investigate the unique role of breastfeeding in establishing infant gut microbiota by comparing the effects of different feeding methods on gut microbiota construction and colonization.

The composition of breast milk changes during breastfeeding to meet the growing infant’s needs [13]. The composition of breast milk varies at each stage, from colostrum to transition milk to mature milk; therefore, there are some differences in microbiota. In this study, it was found that from day 0 to day 30, *Lactobacillus*, *Enterobacter*, *Bifidobacterium*, *Massilia*, *Acinetobacter*, *Pseudomonas*, *Brevundimona*, and *Serratia* in breast milk increased significantly. Previous studies showed that the total bacterial concentration in colostrum was lower than that in transition and mature milk and that the content of bifidobacteria in breast milk during lactation increased over time [14,15], consistent with this study. However, the dominant microbiota species were approximately the same in both periods, with high levels of *Streptococcus*, *Bifidobacterium*, *Staphylococcus*, *Lactobacillus,* and *Rothia*, similar to the 12 major genera of breast milk microbiota found by Murphy [16].

While the breast milk microbiota is dynamic, there is also a dynamic process of building the gut microbiota of infants who use breast milk as their primary food source. In this study, we found some correlations between breast milk and infant gut microbiota. This correlation mainly reflects the possibility that specific microbiota in breast milk may have an indirect effect on the infant gut microbiota. This is mainly due to the selectivity of the infant’s intestine for bacteria. Christopher et al. [17] showed experimentally that at the species level, breastfeeding was significantly associated with 121 different bacterial species, with higher levels of *B. bifidum*, *B. breve*, *B. dentium*, *Lactobacillus*, and *Staphylococcus*. Similarly, our study also showed many correlations; thus, we selected several strains of common beneficial and pathogenic bacteria for further analysis. *Lactobacillus* and *Bifidobacterium*, as the most common beneficial bacteria, can regulate the infant intestinal microbiota, exert antibacterial activity, and inhibit the colonization of pathogenic microorganisms in the intestine [18]. Moreover, they can break down HMOs and promote absorption [19]. Our study selected some of the breast milk microbiota and infant intestinal microbiota separately for correlation analysis, with the aim of exploring which bacteria in breast milk may directly or indirectly affect the infant gut microbiota. Since there were so many species of bacteria that it was not practical to compare the correlation of each, we selected some of the beneficial and pathogenic bacteria with high relative abundance and some representative ones for further correlation analysis. In this study, by observing the correlation of such beneficial strains in breast milk and infants’ feces, we found that in the 0-day BF group, *Lactobacillus* in breast milk suggested a positive correlation with *Bifidobacterium* and *Clostridium* in the infant’s intestine. In contrast, *Lactobacillus* in the infant’s intestine was positively correlated with *Enterobacter* in breast milk. *Lactobacillus* was mainly derived from breast milk for infants. A study showed that *Lactobacillus* was more frequently isolated from infants’ feces receiving breast milk than from weaned infants [20]. Moreover, *Lactobacillus* composition was extremely similar in mother-child pairs [21], meaning that *Lactobacillus* in breast milk influenced the construction of the infant gut microbiota. In the 30-days BF group, *Bifidobacterium* in breast milk was positively correlated with *Alistipes* and *Muribaculaceae* in the infant’s intestine, whereas it was negatively correlated with *Undefined_Enterobacteriaceae*. Existing studies have suggested that *Alistipes* have protective effects against some diseases, including liver fibrosis, colitis, cancer immunotherapy, and cardiovascular disease [22]. Moreover, *Enterobacteriaceae*, as a pathogenic bacterium, may cause various gastrointestinal diseases. Thus, it seems like the beneficial bacteria in breast milk could increase the beneficial bacteria and decrease the harmful bacteria in the infant gut microbiota. In terms of some suspected pathogenic bacteria (e.g., *Escherichia*, *Klebsiella*, and *Clostridium*), we found that in the 0-day BF group, the pathogenic *Klebsiella* in the breast milk was negatively correlated with *Lactobacillus* in the infant’s intestine, whereas *Escherichia-Shigella* was positively correlated with *Ammoniphilus*. The pathogenic bacteria found in the infant intestine, such as *Clostridium* and *Escherichia-Shigella*, suggested a positive correlation with *Blautia*, *Rothia*, and *Parabacteroides* in breast milk. The above results suggest that the pathogenic bacteria in breast milk are harmful to the colonization of beneficial bacteria in the infant’s intestine.

In the 30-days MF group, *Bifidobacterium* in breast milk was positively correlated with *Enterobacter* in the infant’s intestine, whereas *Gemella* and *Massilia* in breast milk were negatively correlated with *Bifidobacterium* in the infant’s intestine. Bifidobacteria in the infant’s intestine are susceptible to various breast milk microbiota, possibly resulting from a more demanding living environment where bifidobacteria are required [23]. Regarding pathogenic bacteria, *Pseudomonas* in breast milk suggested a negative correlation with *Escherichia-Shigella* in the infant’s intestine. Above all, infants in the MF group ingested a wider variety of microbiota, and therefore, more species were correlated with the breast milk microbiota due to the formula introduction [24]. Formula intake provided the bacteria with more available nutrients and altered the bacterial metabolites so that it may increase the correlations. Another theory was that the unique oligosaccharides found in breast milk serve as selective metabolic substrates for a limited number of gut microbes, leading to lower bacterial richness and diversity in the BF group’s infants [19].

Since the formula in the MF group may have altered the bacterial metabolites, we further speculated whether the SCFAs in breast milk indirectly affected the infant gut microbiota. Breast milk contains significant amounts of carbohydrates, which can escape digestion in the small intestine and be broken down into SCFAs by bacteria [7]. Thus, SCFAs in the infant’s intestine are major products of the gut microbiota in the host gut. Recent studies have shown that breastfeeding is closely associated with infant intestinal metabolites and that the infant intestinal metabolites’ SCFAs may be the important mediators of the effects of breastfeeding on the infant gut microbiota. As metabolites of specific fermentable carbohydrates, some SCFAs create anabolic conditions for amino acid-dependent bacterial growth, thereby promoting the growth of such bacteria and regulating the composition of the infant intestinal microbiota [25]. When breast milk is absorbed into the infant’s intestine, breast milk microbiota may change the concentrations and relative ratios of the infant’s SCFAs and intermediate metabolites [7], which may locally affect the infant’s gut microbiota. In this study, we found that the infants’ fecal SCFAs in the BF group were predominantly acetic acid. It is possible that the bifidobacteria in breast milk can ferment and metabolize HMOs through the “Acetyl CoA Pathway” to produce large amounts of acetic acid, which is consistent with the results of Koh [26]. In addition, the number of species correlating between breast milk microbiota and fecal SCFAs in infants was higher in the BF group than in the MF group at both phylum and genus levels. This result further confirmed that the effect of breastfeeding on infant gut microbiota depended more on the mediating effect of SCFAs, whereas the addition of formula in the MF group may provide other influencing substances and thus affect the microbiota’s construction. This study focused on several common types of beneficial and pathogenic bacteria, as their effects on the human body are more obvious. Other microbiota vary greatly among individuals or have not been extensively studied, so we will not discuss them further in this study.

As the infant grows, the composition of the infant’s gut microbiota changes dynamically, with feeding practices being an important factor in the colonization and succession of the infant’s gut microbiota. A study showed that breastfeeding explained the greatest amount of variance in infant gut microbiota until 14 months of life [17]. The effect of breastfeeding on gut microbiota is partly due to the presence of antimicrobial proteins in breast milk (e.g., secretory immunoglobulins, HMOs), which have specific effects on the colonization of certain gut microbiota [25]. Our study concludes that breastfeeding is undoubtedly the best way to feed infants. Breast milk microbiota significantly influences the construction of infant gut microbiota, with some key strains such as *Lactobacillus*, *Bifidobacterium*, and *Enterobacter*. However, the effect of breast milk microbiota on infant gut microbiota is not through direct strain transmission but has been indirectly influenced, which may be related to the cross-feeding effect mediated by SCFAs. The specific mechanism of influence needs further study. This study demonstrates that specific microbiota in breast milk affect the infant’s gut microbiota. Moreover, these results provide a reference for formula supplementation, meaning that formula can be supplemented with beneficial bacteria like those found in breast milk to promote infant gut health. Our study had some limitations. For example, the small sample size and the large individual differences between mothers and infants may be the main reasons influencing this study’s results. In the future, a larger cohort needs to be established for further exploration.

## Figures and Tables

**Figure 1 nutrients-14-05397-f001:**
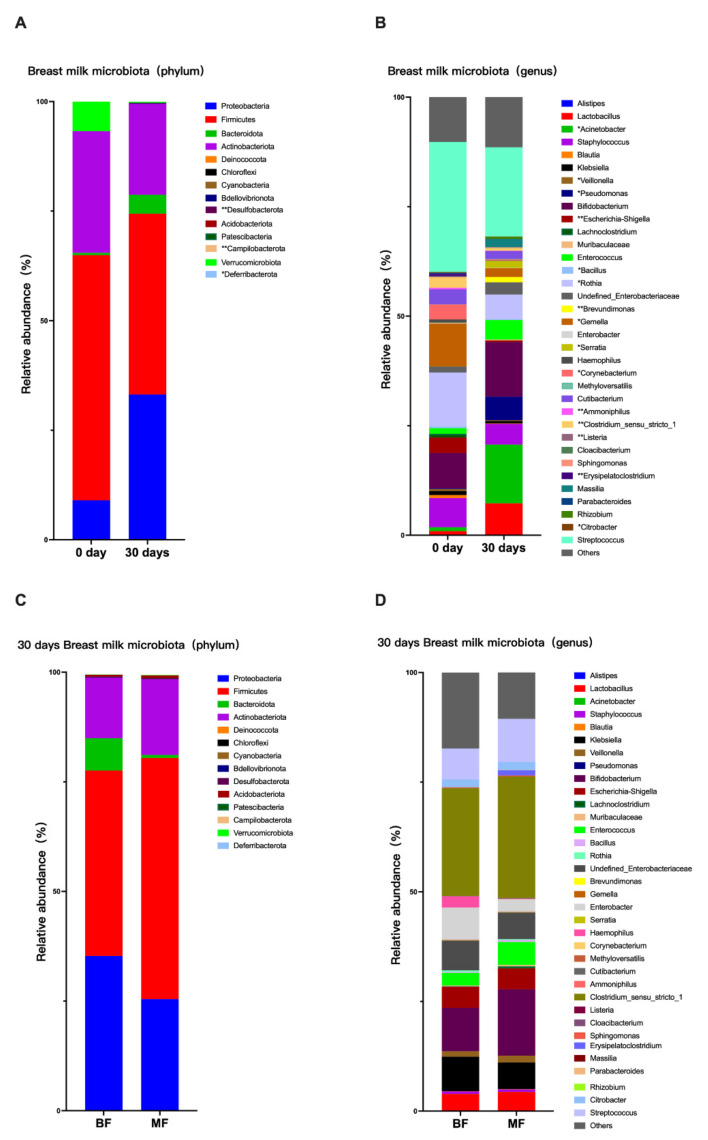
Differences in the relative abundance of breast milk microbiota among different periods and subgroups. (0 day, *n* = 6; 30 days, *n* = 19; BF, *n* = 9; MF, *n* = 10) (**A**) Differences in the relative abundance of breast milk microbiota at 0 day and 30 days at the phylum level; (**B**) Differences in the relative abundance of breast milk microbiota at 0 day and 30 days at the genus level; (**C**) Differences in the relative abundance of breast milk microbiota at the phylum level on 30 days; (**D**) Differences in the relative abundance of breast milk microbiota at the genus level on 30 days. * *p* < 0.05, ** *p* < 0.01.

**Figure 2 nutrients-14-05397-f002:**
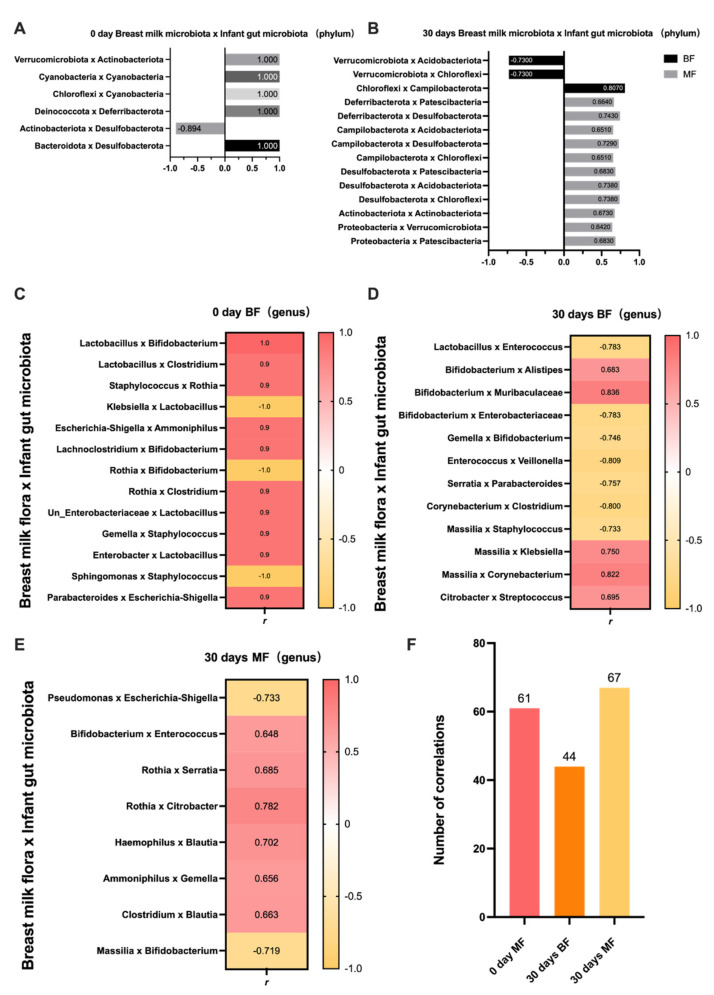
Correlation between breast milk microbiota and an infant’s gut microbiota. (0-day BF, *n* = 5; 30-days BF, *n* = 9; 30-days MF, *n* = 10) (**A**) Correlation between breast milk microbiota and infant’s gut microbiota at the phylum level in the 0-day BF group; (**B**) Correlation between breast milk microbiota and infant’s gut microbiota at the phylum level in the 30-days BF group and 30-days MF group; (**C**) Correlation between breast milk and infant’s gut microbiota at the genus level in the 0-day BF group; (**D**) Correlation between breast milk and infant’s gut microbiota at the genus level in the 30-days BF group; (**E**) Correlation between breast milk and infant’s gut microbiota at the genus level in the 30-days MF group; (**F**) The number of correlations that were significantly different among the 0-day BF group, 30-days BF group, and 30-days MF group.

**Figure 3 nutrients-14-05397-f003:**
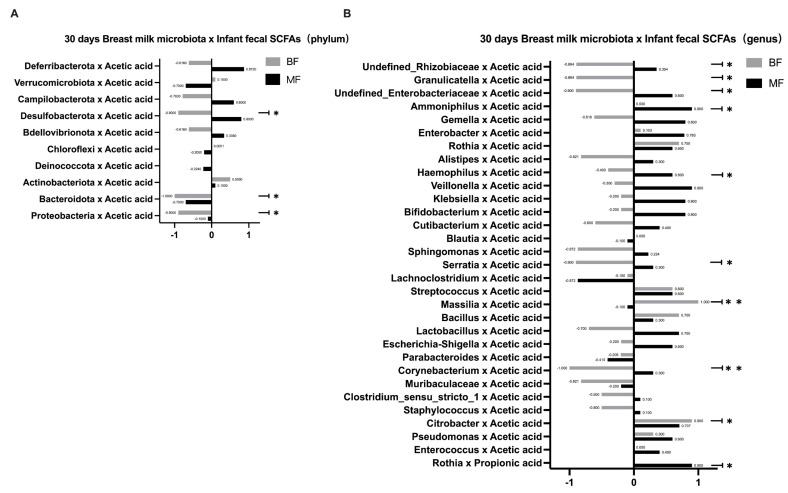
Correlation between breast milk microbiota and SCFAs of infants’ gut microbiota on day 30. (30-days BF, *n* = 5; 30-days MF, *n* = 5) (**A**) Correlation between breast milk microbiota and SCFAs in infants’ feces at the phylum level in the 30-days BF group and 30-days MF group; (**B**) Correlation between breast milk microbiota and SCFAs in infants’ feces at the genus level in the 30-days BF group and 30-days MF group. * *p* < 0.05, ** *p* < 0.01.

**Table 1 nutrients-14-05397-t001:** The sample size and the number of samples in the breastfeeding group (BF) and the mixed feeding group (MF).

	BF		MF	
Subjects *n* = 23	13		10	
	0 day	30 days	0 day	30 days
Breast milk samples	5	9	1	10
Fecal samples	5	9	1	10

**Table 2 nutrients-14-05397-t002:** Comparison of basic information between the breastfeeding group (BF) and the mixed feeding group (MF).

Projects	BF	MF	*t*	*p*
Mother’s age (years)	31.77 ± 2.92	31.50 ± 5.28	0.156	0.877
Pre-pregnancy BMI (kg/m^2^)	20.38 ± 2.27	22.08 ± 2.58	−1.672	0.109
Weight gain during pregnancy (kg)	13.04 ± 5.34	13.16 ± 6.31	−0.05	0.961
Gestational age (weeks)	38.92 ± 0.95	39.20 ± 0.42	−0.853	0.404
Infant length (cm)	48.69 ± 1.89	50.10 ± 1.45	−1.953	0.064
Infant weight (g)	3121.54 ± 409.67	3380.00 ± 311.16	−1.658	0.112

**Table 3 nutrients-14-05397-t003:** Comparison of basic information between the breastfeeding group (BF) and the mixed feeding group (MF).

Projects	BF (People)	MF (People)	*X* ^2^	*p*
Baby Gender				
Male	3	5	1.806	0.179
Female	10	5		
Delivery method				
Nature labor	5	3	0.178	0.673
Cesarean labor	8	7		
Whether probiotics were used during pregnancy (excluding 1 month before delivery)				
Yes	9	10		
No	3	0		0.221 *
Have you used antibiotics during pregnancy (excluding 1 month before delivery)				
Yes	11	10		
No	2	0		0.486 *

* means Fisher’s exact test.

## Data Availability

All data generated or analyzed during this study are included in this published article. If the raw data must be uploaded, the author can provide it.

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
