# Peer review of "The Effect of Breast Milk Microbiota on the Composition of Infant Gut Microbiota: A Cohort Study"

_nutrients, 2022, doi:10.3390/nu14245397_

Round 1
Reviewer 1 Report
This manuscript from Li et al aims to compare the effects of breast milk feeds and mixed BM/formula feeds on the microbial colonization and variety of the infant gut. They conclude that breast milk microbiota partially influence the infant gut microbiota, with some strains having a crucial influence but find that the composition and content of breast milk and infant gut microbiota do not correspond, which may be related to the cross-feeding effect mediated by SCFAs.
The premise behind the studies is certainly interesting however the paper is difficult to follow and several issues must be addressed.
The method by which sequencing has been done is not mentioned.
If n=13 for BM and n=10 for MF subjects, it is not clear how there are 6 0 day BM samples and then 19 samples at 30 days (in methods section). The sample numbers for each group should be clarified.
The authors should present the infant gut microbiota data along with the BM data as a baseline.
How have the groups for correlation been decided? What is the reasoning for the comparisons chosen? E.g., would it not be meaningful to compare the abundance of bifidobacteria in the BM vs stool? More explanation of the methods/reasons for analyses is needed.
Why have all the microbiota been correlated with only acetic acid, except for Rothia?
The Sun Manman reference is not a paper, please clarify the reference.
Reviewer 2 Report
The effect of breast milk microbiota on the composition of infant gut microbiota: A cohort study
Yapeng Li 1,† , Lei Ren 2,† , Yunyi Wang 1 , Jinxing Li 1 , Qingqing Zhou 1,3 , Chenrui Peng 1 , Yuchen Li 4,5 , Ruyue Cheng 1 , Fang He 1 and Xi Shen 1,*
This study is designed to explore the relationship between human milk microbiota and the establishment of infant fecal microbiota during early lactation. The manuscript is well written and easy to read. The results are interesting but some points could be improved or discussed.
- Paragraph 2.2 indicates that 58 fecal samples and 25 breast milk samples were collected. However, paragraph 2.1 mentions a total of 23 newborn and mother pairs included in the study. Can the authors clarify the number of fecal samples collected vs analyzed as well as the distribution of these samples between 0d and 30d?
- The first part of the following sentence of paragraph 2.1 is not clear: "Since there are no infants who do not take breast milk but only formula, the remaining ten infants who were fed formula for ≥ 25 days within 30 days of birth belonged to the MF”
Can the authors clarify this and explain why a "mixed feeding group" and not a formula feeding group was created when it says these infants are taking "formula only"?
- In table 1, some maternal parameters could be simplified by calculating “maternal BMI before pregnancy” and “maternal weight gain”, which could be more indicative in this type of study.
- In Table 1, infant height and weight appear to be higher in the MF group although they are not significant. What can explain this? Is it possible to add the gestational age at delivery in the table?
- Adding a figure presented similarly to Figure 1 for the evolution of infant microbiota composition and group differences would greatly improve the manuscript. Similarly, a figure describing the measured SCFA fecal concentrations would have been appreciated.
- Butyric acid levels in fecal samples have been determined but are not analyzed or at least commented on. Is it possible to add this point in the study?
- Can the author clarify the following sentence in the discussion "In this study, we found correlations between breast milk and infant gut microbiota, but not in composition and content"? What analyzes are they based on?
Minor:
- Figure 2 is too small and difficult to read
- Title of paragraph 2.5: “2.5.16. S rRNA Sequencing and Bioinformatics Analysis »
